# Analysis of the Gálvez–Davison Index for the Forecasting Formation and Evolution of Convective Clouds in the Tropics: Western Cuba

Tahimy Fuentes-Alvarez [1,2,*], Pedro M. González-Jardines [2], José C. Fernández-Alvarez [3,4], Laura de la Torre [3] and Juan A. Añel [3,*]

1   Facultad de Ciencias del Mar, Universidad de Las Palmas de Gran Canaria,
    35017 Las Palmas de Gran Canaria, Spain
2   Centro de Física de la Atmósfera, Instituto de Meteorología, Casablanca, La Habana 10400, Cuba;
    pedro.met90@gmail.com
3   Centro de Investigación Mariña, Universidade de Vigo, Environmental Physics Laboratory (EPhysLab),
    Campus As Lagoas s/n, 32004 Ourense, Spain; jose.carlos.fernandez.alvarez@uvigo.gal (J.C.F.-A.);
    ltr@uvigo.gal (L.d.l.T.)
4   Departamento de Meteorología, Instituto Superior de Tecnologías y Ciencias Aplicadas,
    Universidad de La Habana, La Habana 10400, Cuba
*   Correspondence: thmfuentes@gmail.com (T.F.-A.); j.anhel@uvigo.gal (J.A.A.)

**Abstract:** The Gálvez–Davison Index (GDI) is an atmospheric stability index recently developed to improve the prediction of thunderstorms and shallower types of moist convection in the tropics. Because of its novelty, its use for tropical regions remains largely unexplored. Cuba is a region that suffers extreme weather events, such as tropical storms and hurricanes, some of them worsened by climate change. This research analyzes the effectiveness of the GDI in detecting the potential for convective cloud development, using forecast data from the Weather Research and Forecasting (WRF) model for Western Cuba. To accomplish this, here, we evaluated the performance of the GDI in ten study cases from the dry and wet seasons. As part of our study, we researched how GDI correlates with brightness temperatures (BTs) measured using GOES-16. In addition, the GDI results with the WRF model are compared with results using the Global Forecast System (GFS). Our results show a high correlation between the GDI and BT, concluding that the GDI is a robust tool for forecasting both synoptic and mesoscale convective phenomena over the region studied. In addition, the GDI is able to adequately forecast stability conditions. Finally, the GDI values computed from the WRF model perform much better than those from the GFS, probably because of the greater horizontal resolution in the WRF model.

**Keywords:** GDI; thermodynamics index; convective clouds; brightness temperature; WRF; GFS

## 1. Introduction

Strong convection and humid air in the tropics frequently lead to the development of squalls and thunderstorms. Forecasting storm trajectories is complex because these systems tend to develop and dissipate rapidly, often within an hour or two, and in the tropics, this is because the prevailing winds are also generally weak. This short lifetime and small size make it challenging to produce accurate and timely forecasts.

Most tropical precipitation stems from convective rainfall [1]. Tropical convection is different in many ways from mid-latitude convection. First, latent heat release initiates and fuels convection in the tropics; meanwhile, available potential energy from strong temperature gradients drives convection in the mid-latitudes. Overall, frontal movements and boundaries between air masses resulting from strong temperature gradients primarily cause mid-latitude convection, while large-scale circulations (e.g., the Hadley cell, the

Intertropical Convergence Zone (ITCZ), and the Walker circulation [2,3]) and latent heat release drive convection in the tropics.

In the Cuban archipelago, located within the tropical zone, the predominant climate is the warm tropical type, with a rainy season in the summer [4]. From November to April, subdaily weather variations are more notable than for other months, with sudden changes associated with frontal systems, the anticyclonic influence of continental origin, and low pressure centers from the extratropics. The North Atlantic Subtropical Cyclone influences the weather throughout the year, and from May to October, there are few variations in the weather, as the region remains under the influence of the North Atlantic Subtropical Anticyclone. During June–August, its influence is dominant [5]. It is also known that when the central region of the North Atlantic Subtropical Anticyclone moves significantly away from the Cuban archipelago, there is an extensive displacement of the air masses from their source region, resulting in a more prolonged ocean–atmosphere interaction and elongated isobars, with the wind direction depending on its shape and accompanied by humidity, heat, and upward movements [6].

Although the Cuban archipelago is affected by thunderstorms almost all year, its presence is common in the afternoon mainly during the rainy season (May–October) because of the higher solar radiation that this area receives at this time of year [7], the western region of the archipelago being the one with the highest average number of days with storms [8].

The transition months (April and October) are more favorable for severe phenomena related to rapid convective development. These severe convective phenomena have a considerable socioeconomic impact; the most relevant is death from lightning strikes, the leading cause of death in Cuba associated with natural phenomena, with an annual average of 67 deaths [8]. For their detection, monitoring, and forecast, there are necessary physical–mathematical tools and computational and observational techniques. These usually involve using jointly numerical models and thermodynamic indices computed using their outputs, with high spatial and temporal resolution.

The thermodynamic stability of the atmosphere is critical for the potential and intensity of atmospheric convection, and often, thermodynamic indices are collectively used to assess it. Although stability indices have been widely used to forecast thunderstorms, e.g., [9,10], research has shown that traditional indices (e.g., the Lifted Index (LI), the K-Index (KI), the Showalter Index (SI), or convective available potential energy (CAPE)) often lack capabilities when attempting to predict tropical convection [11].

Previous work has shown that traditional thermodynamic indices are not good at detecting deep convection in the tropics [12] or convective potential in the Cuban archipelago [13–15], making it challenging to quantify tropical convection. This is because, unlike traditional thermodynamic indices, the GDI considers the thermodynamic processes in the boundary layer (better inclusion of the role of latent heat from the ocean) and the inclusion of trade wind inversion [11].

The Gálvez–Davison Index (GDI) was developed by the NOAA Weather Prediction Center (WPC) in 2014, focusing on thermodynamic processes more than on dynamical processes [11] (http://www.wpc.ncep.noaa.gov/international/gdi/, accessed on 13 October 2023) and aiming to improve forecasts for tropical convection. However, since the GDI is a relatively new index, follow-up studies are limited in number.

The GDI's ability to forecast rainfall has been assessed, comparing it with other stability indices (CAPE, KI, and Total Totals) for Puerto Rico [16], finding that the GDI has a much better ability than these other indices. Also, the GDI has been used to analyze long-term trends in convection in the Caribbean region [17], describing the type and potential for precipitation events. Spatial analyses have shown high and statistically significant correlations between the GDI and the annual and seasonal precipitation across the study region.

A similar work examined the effectiveness of both the GDI and KI forecasts using error values across all seasons in an intra-annual study of Northern Africa [18]. The authors

found that the GDI and KI have similar location errors in both intra-annual and intra-seasonal studies. In comparison with the KI, the GDI had lower area error values in the intra-annual study and in most convective synoptic cases with 95% confidence.

The main limitation of the GDI in these studies is related to spatial and temporal resolution. To date, the GDI for the Caribbean region has been calculated based on the Global Forecast System (GFS) [11,17,18] and the Climate Forecast System, version 2 (CFSv2) [16,19], with a horizontal resolution greater or equal to 1° and a temporal resolution of 6 h or greater. Such resolutions are greater than what is desirable for forecasting mesoscale phenomena.

In this work, we selected two models with different spatial and temporal resolutions to compare the ability of the GDI to detect convective potential in Western Cuba using the GFS and the Weather Research and Forecast (WRF) model. It is expected that the WRF model provides greater detail and more accurately represents the intensity and spatial location of smaller-scale weather events compared with the GFS [20]. Therefore, in this work, the GDI is calculated with a greater spatial and temporal resolution using data from the WRF model, testing if the GDI is a suitable tool for forecasting convection over Cuba, and if, as we expect, a priori, it improves the forecasting ability of the GDI.

Another characteristic of the GDI is that, given its relative novelty, only very few studies have used it or researched its performance and validity (e.g., [11,16–18,21–24]). Therefore, in this study, we tested the ability of the GDI to analyze the formation and evolution of convective clouds over Western Cuba during different seasons. Also, to date, the study of convective episodes over Western Cuba using thermodynamic indices shows that these are not an effective tool for convection forecasts [13–15], and given the frequent occurrence of severe local storms in this region [25], it is necessary to research and incorporate new tools to improve forecasts of convective potential. Therefore, here, we used the GDI index to evaluate its ability to forecast convective potential in Western Cuba.

The remainder of the paper is structured such that Section 2 presents the region of study and study cases selected and the model and satellite data used, followed by a description of the index evaluation. The results and their discussion are shown in Section 3, with an evaluation of GDI forecasts based on satellite brightness temperature values and a comparison of the results obtained with the WRF and GFS models. Finally, the conclusions are presented in Section 4.

## 2. Materials and Methods

### 2.1. Region of Study

The western region of Cuba was selected as the study area (Figure 1) because it is highly affected by storms during most of the year. In this region, the average temperatures are around 25 °C, and the relative humidity is generally high, with values ranging between 75% and 95%. Also, this region is one of the three different zones for precipitation regimes in Cuba, according to previous studies [26]. The rainy season occurs from May to October and is especially pronounced in September and October; sea breezes prevail on both the north and south coasts. Their convergence, together with daytime warming, favors the occurrence of thunderstorms in inland locations during the afternoon hours. On the other hand, the dry season, from November to April, is the coolest and is characterized by the influence of extratropical frontal systems.

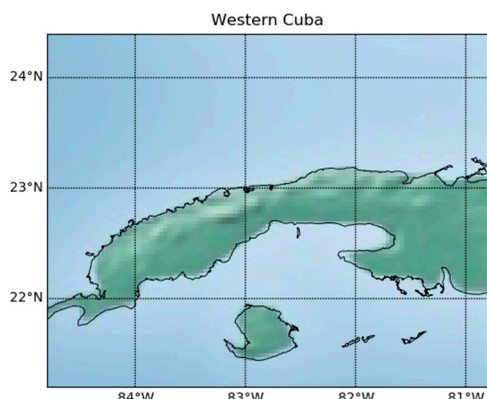

**Figure 1.** Region of study: Western Cuba (21.2–24.4° N, 80.8–84.8° W).

*2.2. Study Cases*

To perform our study, we chose ten representative cases over the period 2017–2019, as we consider them a large enough sample to test the hypotheses of this work (see Table 1). The cases are classified into the four most frequent types of meteorological conditions in the region: summer storms, prefrontal storm lines, hurricanes, and stable conditions. These include different types of phenomena during the rainy and dry periods of the year, and sometimes, these summer storms may be accompanied by severe events and are classified as severe local storms. This definition is taken following the literature on the classification of these phenomena in Cuba [27], which traditionally refers to those storms that present at least one of the following phenomena: waterspouts, tornadoes, hail, and linear winds greater than 96 km/h. In addition, we evaluated the behavior of the GDI under stable conditions.

**Table 1.** Case studies. The types refer to the different meteorological conditions that most frequently generate significant convective activity in Western Cuba (types 1, 2, and 3). Type 4 represents conditions of atmospheric stability.

| Date | Type | Meteorological Condition |
|---|---|---|
| 16 August 2017 | 1 | Summer storms |
| 18 August 2017 | 1 | Summer storms |
| 21 August 2017 | 1 | Summer storms |
| 9 September 2017 | 2 | Hurricane Irma |
| 25 October 2017 | 3 | Prefrontal storms line |
| 8 October 2018 | 2 | Hurricane Michael |
| 21 December 2018 | 3 | Prefrontal storms line |
| 28 January 2019 | 3 | Prefrontal storms line |
| 7 March 2019 | 4 | Stability |
| 8 March 2019 | 4 | Stability |

*2.3. Data*

To calculate the GDI, the values of temperatures and mixing ratios are used from four levels: 950, 850, 700, and 500 hPa, which are used to define three layers. Layer A (data at the fixed level of 950 hPa) represents thermodynamic conditions in the boundary layer. Layer B captures the variability associated with the trade wind inversion (TWI) [28] using a simple average of 850 and 700 hPa data. Layer C represents the mid-troposphere by considering the data at a fixed level of 500 hPa. A summary of GDI values and their associated convective regimes is shown in Table 2 [11]. The National Oceanic and Atmospheric Administration

provides a complete methodology to calculate the GDI and details on validation (https: //www.wpc.ncep.noaa.gov/international/gdi/, accessed on 13 October 2023).

**Table 2.** GDI scale according to the expected convective potential [11].

| GDI Value | Expected Convective Regime |
| --- | --- |
| GDI > +45 | Scattered-to-widespread heavy rain-producing thunderstorms. |
| +35 to +45 | Scattered thunderstorms, some capable of producing heavy rainfall. |
| +25 to +35 | Scattered thunderstorms or scattered shallow convection with isolated thunderstorms. |
| +15 to +25 | Isolated thunderstorms but mostly shallow convection. |
| +5 to +15 | Shallow convection. A very isolated and brief thunderstorm is possible. |
| −20 to +5 | Isolated to scattered shallow convection. Strong subsidence inversion likely. |
| −20 > GDI | Strong subsidence inversion. Convection should be very shallow, isolated, and produce trace accumulations. |

The GDI forecasts are calculated using the temperature (T) and mixing ratio (r) data from the GFS v14.0 (https://www.emc.ncep.noaa.gov/emc/pages/numerical_forecast_ systems/gfs.php, accessed on 13 October 2023) and WRF v3.9 [29] model outputs. The GFS data are available from https://nomads.ncdc.noaa.gov/data/gfs4. GFS is currently used by the National Centers for Environmental Prediction (NCEP) of NOAA to forecast the GDI with a horizontal resolution of 0.25° (approximately 54 km).

The WRF simulations are performed with the dynamical core WRF-ARW (used for operational forecast) and run in the Cuban Meteorology Institute (INSMET) computational infrastructure. Two nested grids are used: the external domain was built with a horizontal resolution of 12 km and time intervals of 3 h, while the nested domain was designed with a horizontal resolution of 4 km and a time interval of one hour (Figure 2). The microphysics scheme used is the WRF Single-Moment-5-class [30], with Grell–Freitas [31,32] convection parameterization and Mellor–Yamada–Janjic [33] as the scheme for the boundary layer. The initial and boundary conditions for the WRF-ARW simulations are forecast data from the GFS, with a horizontal resolution of 0.5° and time intervals of 6 h, starting at 00:00 UTC, and with a 54 h forecast. This model was validated in a previous work for the José Martí International Airport in Havana, which is located within the analyzed study region [34].

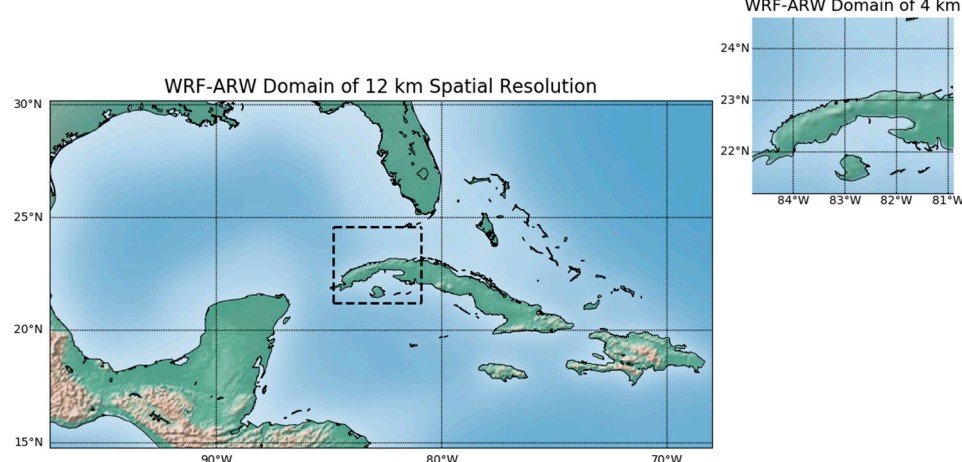

**Figure 2.** Domain used for simulations. The external domain (14.8–30.2° N, 68–97.5° W) and the nested domain (21.2–24.4° N, 80.8–84.8° W).

In addition, the Python programming language [35] was used for processing the data and obtaining the results.

### 2.4. Evaluation

To determine the ability of the GDI to forecast the formation and evolution of convective clouds, we used the correlation between the GDI forecasts and the brightness temperature (BT) from the GOES-16 satellite. We chose the BT because it represents the general structure and depth of the tropical convection [11], and it has proved to be a good proxy to distinguish between different rainfall and cloud types [36]. The BT data were provided by the Center for High-Performance Computing—Research Computing and Data Support of the University of Utah (http://home.chpc.utah.edu/~u0553130, accessed on 13 October 2023), with a spatial resolution of 2 km. To compare the BT and the GDI values, we interpolated the BT data into the GDI grid. We tested the nearest-neighbor, cubic, and spline interpolation methods (not shown here), and the results are similar for all of them because of the high horizontal resolution of the BT data. This result is in accordance with the literature [37], and finally, the nearest neighbor interpolation method was used. The correlation between GDI and BT was calculated for both models.

The determination coefficient (DC) is calculated based on the Pearson correlation coefficient:

$$r = \frac{N\Sigma XY - (\Sigma X)(\Sigma Y)}{\sqrt{N(\Sigma X^2) - (\Sigma X)^2}\sqrt{N(\Sigma Y^2) - (\Sigma Y)^2}} \tag{1}$$

where $X$ corresponds to the values of the GDI, $Y$ is the BT values, and $N$ is the size of the sample.

## 3. Results and Discussion

The results obtained are shown by the type of meteorological condition: summer storms (type 1), hurricanes (type 2), prefrontal storm lines (type 3), and stability (type 4).

### 3.1. GDI Evaluation Using the BT

#### 3.1.1. Summer Storms

Three of the case studies here happened in an interval of only five days in August 2017. On the 16th, 18th, and 21st, there was significant convective activity toward inland areas of the study region, becoming strong in some locations. There were reports of severe local storms from some weather stations. The highest reflectivity observed by the weather radar ranged between 50 and 55 dBZ on 16 August (see Figure 3 left), which qualifies as a strong storm.

For the summer storm cases analyzed, the GDI forecast correctly represents the most favorable regions for the storms to break out, fundamentally toward the western inland (Figure 3 right). This indicates the possibility of deep convection with GDI values above 45, which happened. The GDI shows regions with the potential to develop strong convection, which did not occur. In this case, the GDI indicates that the potential for convective development exists at those points; however, it may be that the potential exists and convective development does not occur because of other mesoscale factors.

#### 3.1.2. Hurricanes

Regarding tropical cyclones, the GDI forecast shows the different parts of a hurricane (the rainbands, the eye, and the eyewall) for both cases. Figure 4 shows it for the case of Hurricane Michael on 8 October 2018. This was an intense hurricane with maximum sustained winds over the study region around 130 km/h and intense rainfall with accumulated amounts equaling 214 mm [38], in addition to all the unaccounted socioeconomic impacts (e.g., agriculture, houses, etc.).

In Figure 4, it is possible to see areas of maximum convective potential associated with the eyewall with values greater than 60, areas with values between 35 and 45 correspond-

ing to the spiral bands, and values between 45 and 55 associated with deep convection embedded within these bands. In addition, it could show regions with values lower than 30 corresponding to the subsiding movements of the systems.

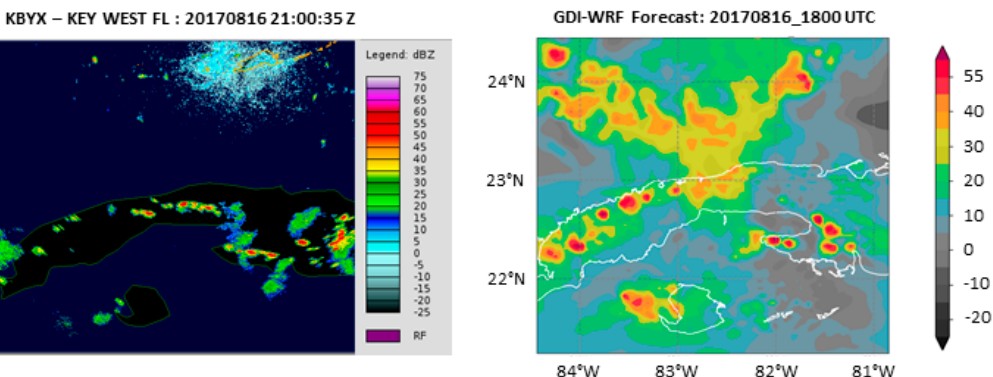

**Figure 3.** (**Left**) Observation from the Key West Radar on 16 August 2017 (21:00 UTC). The reflectivity values between 50 and 55 dBZ (red) indicate the occurrence of strong storms, with the possibility of severe weather. (**Right**) GDI forecast from the WRF model on 16 August 2017 (18:00 UTC). The GDI values greater than 45 (red) show areas with the potential for scattered-to-widespread heavy rain and thunderstorms. Orange and yellow colors show areas with potential for scattered thunderstorms or scattered shallow convection, some capable of producing heavy rainfall and thunderstorms, and areas with GDI values lower than 25 correspond to the potential for a few isolated thunderstorms, although probably mostly shallow convection. The blue ones indicate that a very isolated and brief thunderstorm is possible, and the gray ones show subsidence movements.

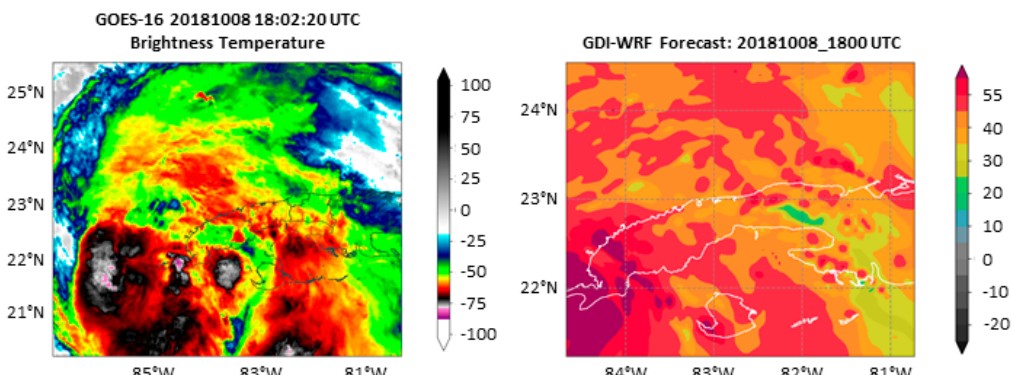

**Figure 4.** Hurricane Michael. (**Left**) Brightness temperature from the GOES-16 ABI Band 14 satellite on 8 October 2018 (18:02 UTC). The cloudiness with the lowest brightness temperature, and, therefore, the deepest, is represented by values below 55 (yellow, red, and purple); higher values represent clouds with less convective but significant development (green and blue), and white areas represent less activity. (**Right**) GDI forecast from the WRF model for 8 October 2018 (18:00 UTC). The GDI values greater than 45 (red and purple) show areas with the potential for scattered-to-widespread heavy rain, producing thunderstorms.

Hurricane Irma (case not shown) was reported as one of the most extreme precipitation events affecting Cuba since 1980 [26], with maximum accumulated precipitation of 225 mm in the northern part of the study region. The maximum sustained winds were higher than 295 km/h, and it was the worst coastal flooding to date, with wave heights between 8 and 10 m [39]. In this case, similar results to Michael were obtained, with a limitation: the translation movement of the hurricane was not well forecasted, and although the precipitation and cloudiness matched with reality, the bands of cloudiness associated with it reached a geographical location one hour later than expected.

### 3.1.3. Prefrontal Storm Lines

For the three cases of prefrontal storm lines (cold fronts) studied, the GDI forecast depicts the classic carrot pattern that characterizes this meteorological phenomenon (e.g., Figure 5). Of the cases analyzed, this was the one selected to show due its impact in the study region. One of the storms embedded in this band evolved into an intense tornado (category EF4), which caused six deaths and injured many people.

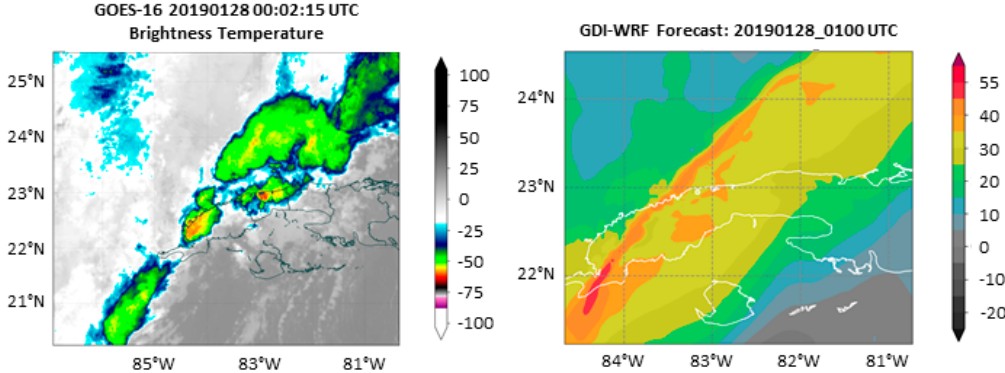

**Figure 5.** Prefrontal storm line: (**left**) BT from the GOES-16 ABI Band 14 satellite on 28 January 2019 (00:02 UTC).

In these cases, the GDI presents mostly values between 35 and 55, suggesting that there is potential for the development and evolution of storms associated with deep convection in these areas. This is verified with satellite images and radar observations.

The cloudiness with the lowest BT, and, therefore, the deepest, is represented by values below 55 (yellow and red); higher values represent clouds with less convective but significant development (green and blue), and white areas represent less activity or nothing. The right figure depicts the GDI forecast from the WRF model for 28 January 2019 (01:00 UTC). GDI values greater than 45 show areas with the potential for scattered-to-widespread heavy rain, producing thunderstorms (red).

In general, the GDI reflects the evolution of the phenomenon very well. The areas of lower BT correspond with maximum GDI values. It detects the advance of the prefrontal storm line, a region of less convective activity before it, strong storms in the front line, and a region of strong subsidence behind the front line. A limitation of the GDI-WRF forecast is that the translation movement of the prefrontal line is one hour later than its actual position.

### 3.1.4. Stability

We evaluated the GDI for two anticyclonic cases. For these, the GDI presents values ranging between −20 and 5, indicating stability conditions and low storm probability (Figure 6, right). It only shows some areas with values between 5 and 10 toward the inland and southern areas, which means that, in the event of a storm, it would be brief and isolated, related to the transport of moisture from the sea, together with daytime heating in the afternoon.

### 3.2. Correlation between the GDI Forecast and BT

The GDI-WRF model for the convection days associated with mesoscale conditions fundamentally presents DC values higher than 0.6 toward the interior and south of the region (Figure 7a). These regions coincide with the areas of greatest convective cloudiness. However, when calculating the DC for GDI-GFS (Figure 7b), the values are less than 0.5. This confirms that GDI-GFS is not able to detect significant mesoscale details such as convective storm occurrence.

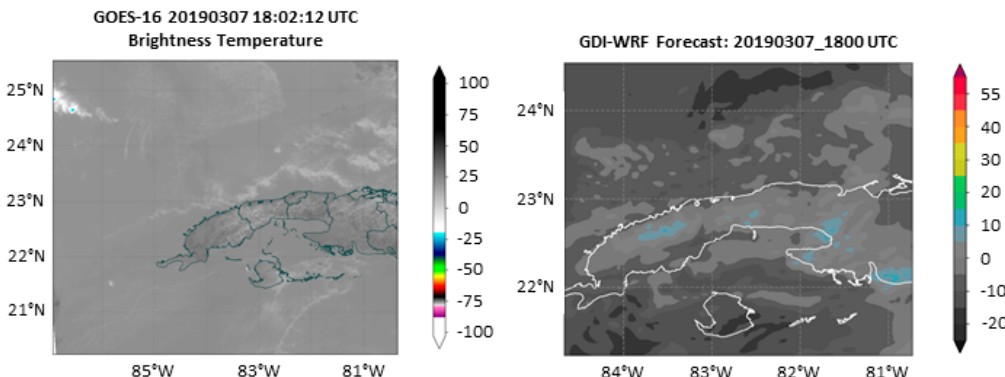

**Figure 6.** Stability conditions: (**left**) Brightness temperature from the GOES-16 ABI Band 14 satellite on 7 March 2019 (18:02 UTC). Values above −20 represent relatively high-temperature clouds (white and greay). (**right**) GDI forecast from the WRF model for 7 March 2019 (18:00 UTC). Values between 5 and 10 (blue) indicate that a very isolated and brief thunderstorm is possible, and gray ones show strong subsidence inversion.

The correlations for the cases of tropical cyclones are higher than 0.5 in the areas of most significant convection, only showing values greater than 0.5 toward the central dense overcast (CDO) and some regions of the spiral bands (Figure 7c). For GFS (Figure 7d), the correlations are weak for most of the domains. In these cases, the BT values do not represent the deep convection because of cirrus contamination.

In cases of frontal systems, the GDI-WRF model presents strong correlations with BT, with DC values greater than 0.75 in a large part of the study region (Figure 7e). The correlation plots for the GDI calculated from the GFS (Figure 7f) present high correlations with DC values greater than 0.5 in a large part of the domain. However, these correlations are weaker than those obtained for the GDI-WRF.

For the stability cases, the DC values corresponding to the GDI-WRF model (Figure 7g) are greater than 0.5 toward some regions of the interior and north of the study area. In these, the convergence of north and south breezes occurs, giving rise to the formation of clouds with little vertical development in stability conditions. The DC values, when calculating the index from the GFS, are less than 0.5 in both cases (Figure 7h). This relates to the deficiencies the GFS model presents when identifying mesoscale details.

### 3.3. GDI-WRF vs. GDI-GFS Forecast

Here, to compare the ability of the GDI calculated from the WRF model and GFS, we split the analysis again according to the event type. To facilitate the comparisons, in the remainder of the manuscript, we refer to the GDI-WRF and GDI-GFS forecasts as the GDI calculated from the WRF and GFS data, respectively.

For type 1 cases, the GDI-GFS forecasts present a lower level of detail because the model could not forecast small areas affected by storms. The GDI shows values below 35, so the strong convective activity that occurred was not predicted. Figure 8a corresponds to the GDI-GFS forecast for 16 August 2017. We can see how the GDI-GFS is not able to capture the mesoscale details obtained with the GDI-WRF model (Figure 8b). The GDI values remained below 35, and, therefore, it did not forecast a high potential for thunderstorm development. As a result, the areas with the potential for developing dangerous phenomena, which the meteorological stations reported in the end, were not predicted by the GDI-GFS.

For type 2 cases, the GDI-GFS forecast shows very similar values in the whole domain, probably because of the coarse resolution of the horizontal grid of the model. This is a downside of the forecast because the important details of hurricanes were lost in the two cases analyzed, such as the morphology of the system, the subsidence zones between the spiral bands, and the different convective regimes present in these bands. Instead, the GDI-WRF forecast shows the different parts of the hurricane. An example can be seen in Figure 8c,d, which shows the GDI-GFS and GDI-WRF forecasts for Hurricane Michael.

For type 3 cases, we obtained a representation of the frontal system and its different expected convective regimes (an example is shown in Figure 8e,f for both models). The GDI-GFS detects the advance of the prefrontal storm line, a region of less convective activity before it, storms in the front line, and a region of strong subsidence behind the front line. This model overestimates the size of the areas with a higher potential for the occurrence of deep convection in the storm line. Similarly, the predicted translation movement of the prefrontal line is one hour later than its actual position.

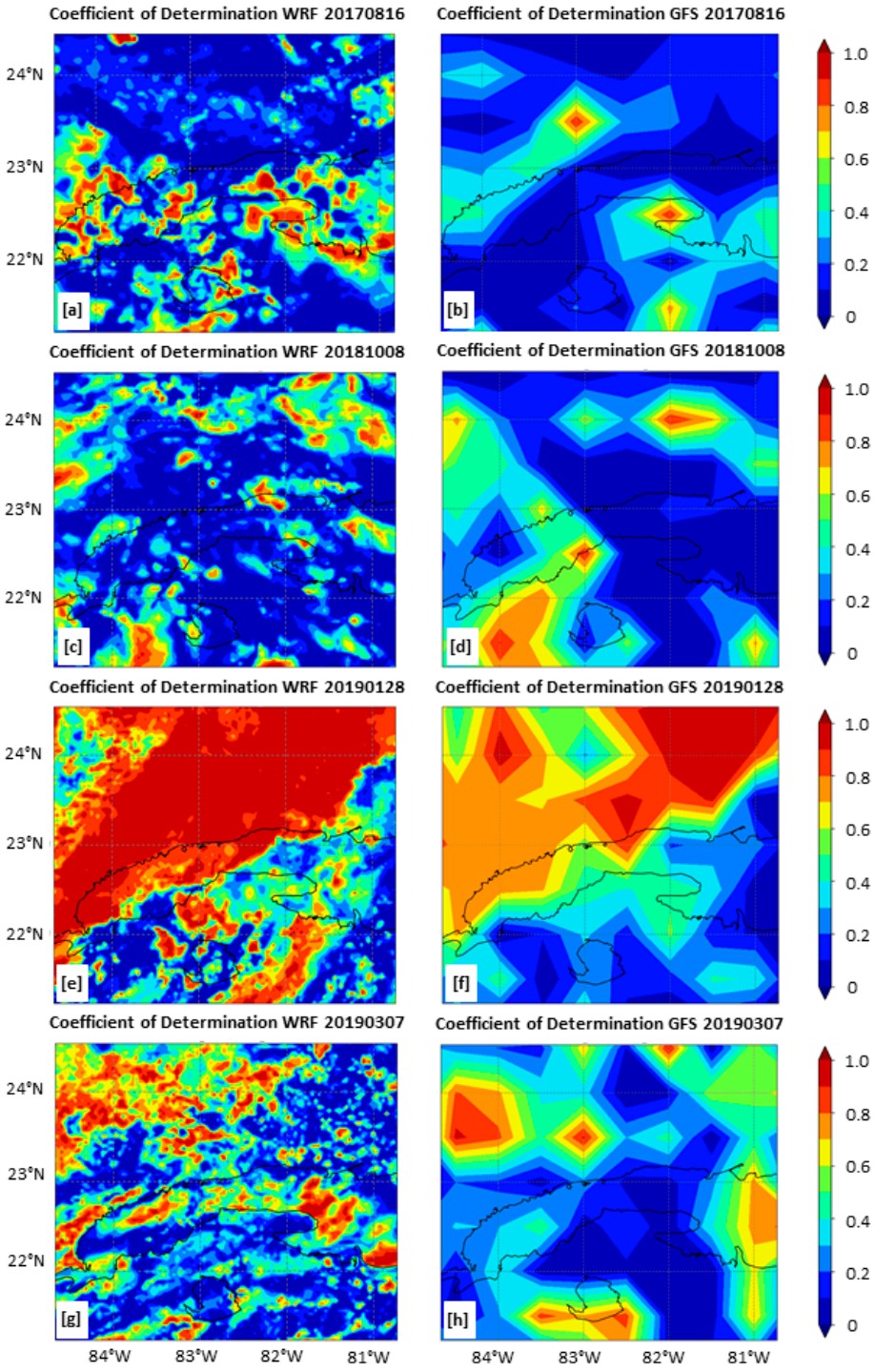

**Figure 7.** Determination coefficient for a 24 h forecast: correlation between GDI-WRF and BT (left column) and GDI-GFS and BT (right column) (**a**,**b**) for August 16, 2017; (**c**,**d**) October 8, 2018; (**e**,**f**) 28 January 2019; and (**g**,**h**) 7 March 2019.

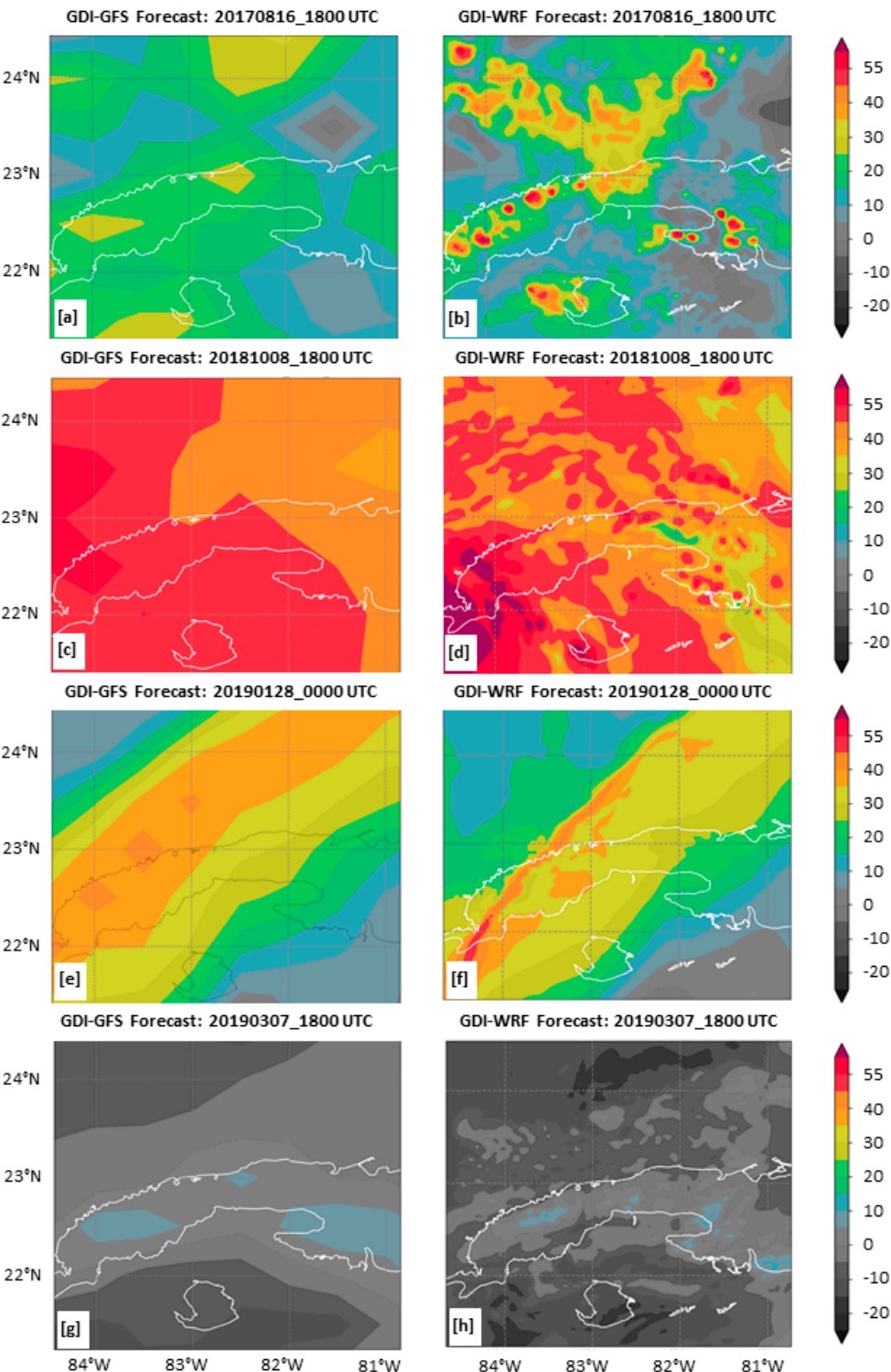

**Figure 8.** GDI-GFS forecast (left column) and GDI-WRF forecast (right column) (**a,b**) for 16 August 2017; (**c,d**) 8 October 2018; (**e,f**) 28 January 2019; and (**g,h**) 7 March 2019.The values greater than 45 show areas with the potential for scattered-to-widespread heavy rain-producing thunderstorms (red). Some areas have the potential for scattered thunderstorms or scattered shallow convection, some are capable of producing heavy rainfall and thunderstorms (orange and yellow), and areas with GDI values lower than 25 have the potential for few isolated thunderstorms, although they mostly produce shallow convection. The blue ones indicate that a very isolated and brief thunderstorm is possible, and the gray ones show subsidence movements.

For type 4 cases (stable conditions), the GDI-GFS forecasts show very generalized contours (an example is shown in Figure 8g), and the values are very homogeneous in the whole domain, overestimating the areas with values between 5 and 15 toward the interior and south.

## 4. Conclusions

In this work, we analyzed the ability of the GDI to forecast convective precipitation in Western Cuba. Also, we tested if using the WRF model instead of the GFS and an increased resolution improves the forecast. Our results show that the GDI has a very good forecast performance for several different case studies, which agrees with previous results in the literature (e.g., [16]).

We found that implementing the GDI-WRF model allows for the analysis of the formation and evolution of convective clouds with a more remarkable ability to forecast the areas with the potential for the convective development of mesoscale phenomena compared with the GDI-GFS model. For synoptic scale phenomena (types 2 and 3), a limitation of the GDI-WRF and GDI-GFS forecasts is that the translation movement is one hour ahead or later than their actual position. For frontal systems, the GDI-WRF and GDI-GFS forecasts show values for the determination coefficient higher than 0.6 in the regions of convective development. For this type of phenomenon, either of the two models can be used, but for greater detail in the expected convection, the GDI-WRF forecast is more accurate. For hurricanes, the GDI-WRF forecast is able to show the different parts of the system; the GDI-GFS forecast lacks this capability. For local-scale phenomena and stability conditions, it is necessary to use the GDI-WRF forecast given the higher spatial resolution.

We found that the GDI-WRF forecast outperforms the GDI-GFS forecast regarding the correlation with BT, with determination coefficient values greater than 0.6 between BT and the GDI-WRF model in the first 24 h of the forecast versus less than 0.4 for most of the analyzed domain in the GDI-GFS model, highlighting the greatest differences in cases of smaller scale.

Previous work reported room for improvement in the GDI for forecasting precipitation in the near region of Puerto Rico [16], stating that the GDI, a priori, was not able to capture cold fronts, troughs, or trade winds' orographic precipitation; however, these are common conditions within the region that we studied here, and the GDI works well. Indeed, the DC values obtained by [11] for the region studied here were lower than 0.4 with the GDI-GFS model, while we obtained values higher than 0.6 with the GDI-WRF model. Also, recent work on the estimation of extreme precipitation using the WRF model and the GDI for the tropical country of El Salvador found improvements in forecasting extreme precipitation using the GDI based on the WRF model and an ensemble of GFS simulations (GEFSs) when compared with GEFSs alone [40]. Therefore, we speculate that the resolution of the model plays a critical role in such differences, and this could present an opportunity to improve the GDI's abilities.

Overall, the use of the GDI enables an improved forecast of convective rainfall for Western Cuba, a notable advance that can help to prevent some of the mentioned risks (deaths, socioeconomic, etc.) in the region associated with these phenomena, as the Caribbean is highly vulnerable to extreme rainfall under climate change [41]. Using the WRF model also represents a notable advancement over the use of the GFS; however, no doubt a good part of such improvement comes from the fact that the WRF model is run at the INSMET for operational purposes with a greater spatial resolution than the GFS, a feature already reported [11].

**Author Contributions:** Conceptualization, T.F.-A. and P.M.G.-J.; methodology, T.F.-A. and P.M.G.-J.; software, T.F.-A., P.M.G.-J. and J.C.F.-A.; validation, T.F.-A., P.M.G.-J. and J.C.F.-A.; formal analysis, T.F.-A. and P.M.G.-J.; investigation, all authors; resources, P.M.G.-J., J.A.A. and L.d.l.T.; data curation, T.F.-A., P.M.G.-J. and J.A.A.; writing—original draft preparation, T.F.-A.; writing—review and editing, all authors; visualization, T.F.-A., P.M.G.-J. and J.A.A.; supervision, P.M.G.-J. and J.A.A.; project

administration, P.M.G.-J., J.A.A. and L.d.l.T.; funding acquisition, P.M.G.-J., J.A.A. and L.d.l.T. All authors have read and agreed to the published version of the manuscript.

**Funding:** T.F.A. was supported in her MSc studies by a scholarship from the Fundación Carolina during the last stages of the development of this work. The EPhysLab is funded by the Xunta de Galicia, Spain, under grant ED431C 2021/44, "Programa de Consolidación e Estructuración de Unidades de Investigación Competitivas" (Grupos de Referencia Competitiva).

**Data Availability Statement:** Because of severe limitations in the IT infrastructure of the INSMET, it was not possible to save the output files used in this work.

**Acknowledgments:** This work is dedicated to Carlos M. González-Ramírez, who participated in its conceptualization and recently passed away. The authors would also like to thank the Cuban Meteorological Institute (INSMET) and the Higher Institute of Technologies and Applied Sciences (InSTEC) of the University of Havana for the support provided during the realization of this work. Also, we would like to thank Francisco J. Tapiador from the Universidad de Castilla-La Mancha for comments on early versions of this manuscript and J.C.F.-A. acknowledge the support from the Xunta de Galicia under the grant no. ED481A-2020/193.

**Conflicts of Interest:** The authors declare no conflict of interest.

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
