# Peer review of "Analysis of the Gálvez–Davison Index for the Forecasting Formation and Evolution of Convective Clouds in the Tropics: Western Cuba"

_climate, doi:10.3390/cli11100209_

Round 1

Reviewer 1 Report

General comments:

This work evaluates the efficiency of the GDI using two models, WRF and GDI, to forecast convective phenomena over Wester Cuba.

This work is well-documented, and the methodology has been performed with clarity. 

Before publication, a brief discussion about the selection of both models is necessary, i.e., GFS has been designed to operate with less resolution, different data assimilation, etc. Then, the lack of accuracy regarding the WRF results was expected. 

Please also explain why BT instead of OLR. 

Specific comments:

L 16 How accurate is this affirmation since it has not been documented in this work?  

L 47-48 Please elaborate on the influence of the North Atlantic Anticyclone over Cuba. How does it contribute to the rainy season in summer?

L 257 The time of the images is interchanged in Figure 3 caption.

L 344 CD has not been defined before.

L 353 TB has not been defined before.

Reviewer 2 Report

This paper is focused on the formation and evolution of convective clouds in tropics. The region investigated is located in Western Cuba, and the parameter used is the Gálvez-Davison Index (GDI). Ten cases were used, and a comparison between the brightness temperature and the GDI was considered. Finally, two models, WRF and GFS, were employed to forecast the thunderstorms. The main feature of the paper is the lack of experience with this index, since it was recently developed. Consequently, the paper may be published in Climate after the introduction of the following minor changes.

In its current form, the paper is a qualitative analysis of the results. Perhaps, the paper could be improved with some quantitative information, such as the surface affected by the storms or the quantitative comparison between values observed and/or modelled.

However, the main inconvenience is the lack of references for the discussion. The paper results are presented but they are not compared with those from other studies. The reader could think that the paper is isolated and not connected with the current research lines focused on tropical storms. The comparison with papers in the same field is useful to place this research in its context.
